# Identification of Genetic Variants Associated with Severe Myocardial Bridging through Whole-Exome Sequencing

**DOI:** 10.3390/jpm13101509

**Published:** 2023-10-18

**Authors:** Tsung-Lin Yang, Jafit Ting, Min-Rou Lin, Wei-Chiao Chang, Chun-Ming Shih

**Affiliations:** 1Graduate Institute of Clinical Medicine, College of Medicine, Taipei Medical University, Taipei 11031, Taiwan; 151017@h.tmu.edu.tw; 2Department of Internal Medicine, School of Medicine, College of Medicine, Taipei Medical University, Taipei 11031, Taiwan; 3Division of Cardiology, Department of Internal Medicine, Taipei Medical University Hospital, Taipei 11031, Taiwan; 4Cardiovascular Research Center, Taipei Medical University Hospital, Taipei 11031, Taiwan; 5Taipei Heart Institute, Taipei Medical University, Taipei 11031, Taiwan; 6Department of Clinical Pharmacy, School of Pharmacy, College of Pharmacy, Taipei Medical University, Taipei 11031, Taiwan; jkt5265@tmu.edu.tw (J.T.); jennlin@tmu.edu.tw (M.-R.L.); 7Master’ Program in Clinical Genomics and Proteomics, School of Pharmacy, Taipei Medical University, Taipei 11031, Taiwan; 8Integrative Research Center for Critical Care, Wan Fang Hospital, Taipei Medical University, Taipei 11696, Taiwan

**Keywords:** myocardial bridging, cardiovascular disease, genetic variant, whole-exome sequencing, *DMD*, *SGCA*, *TTN*

## Abstract

Myocardial bridging (MB) is a congenital coronary artery anomaly and an important cause of angina. The genetic basis of MB is currently unknown. This study used a whole-exome sequencing technique and analyzed genotypic differences. Eight coronary angiography-confirmed cases of severe MB and eight age- and sex-matched control patients were investigated. In total, 139 rare variants that are potentially pathogenic for severe MB were identified in 132 genes. Genes with multiple rare variants or co-predicted by ClinVar and CADD/REVEL for severe MB were collected, from which heart-specific genes were selected under the guidance of tissue expression levels. Functional annotation indicated significant genetic associations with abnormal skeletal muscle mass, cardiomyopathies, and transmembrane ion channels. Candidate genes were reviewed regarding the functions and locations of each individual gene product. Among the gene candidates for severe MB, rare variants in *DMD*, *SGCA,* and *TTN* were determined to be the most crucial. The results suggest that altered anchoring proteins on the cell membrane and intracellular sarcomere unit of cardiomyocytes play a role in the development of the missed trajectory of coronary vessels. Additional studies are required to support the diagnostic application of cardiac sarcoglycan and dystroglycan complexes in patients with severe MB.

## 1. Introduction

The supply of blood to the heart depends on coronary arteries, which are normally distributed in the epicardial space on the surface of the heart. The diastolic phase accounts for most blood flow perfusion to the coronary arteries, meeting the metabolic demands of the heart. Myocardial bridging (MB) is a coronary vascular anomaly where coronary vessels are embedded into heart muscles, making a “bridge” of the muscle band above the feeding vessel, resulting in vascular compression during heart contractions (Figure 1) [1,2]. Because MB squeezes the luminal tunnel of the affected vessel and hampers coronary blood flow, the distal blood perfusion of cardiac tissues would be compromised. Consequently, chest tightness or exertional chest pain may occur, whenever strenuous cardiac muscle contraction is induced, mimicking atherosclerotic coronary artery disease (CAD). Although MB was reported to alleviate atherosclerotic processes of the affected vessel, the proximal end of the MB segment has a higher probability of atherosclerosis due to blood stasis, focal turbulent flow, and shear-stress-related endothelial damage [3,4]. MB causes myocardial ischemia, and the subsequent hemodynamic consequence could not be ignored due to ischemic cardiomyopathy [5]. In a meta-analysis, MB was shown to be associated with an increased risk of adverse cardiac events, non-fatal myocardial infarction, and hospitalization for chest pain [6]. Clinically, the spectrum of symptoms from MB spans extensively. MB can be asymptomatic or manifest as minor angina to severe chest pain [7]. In some rare cases, major cardiac events or life-threatening ventricular arrhythmias may occur during extreme exercise [8,9,10]. The presence of MB was reported to increase the risk of fatal arrhythmia among those with hypertrophic cardiomyopathy and those who underwent implantable cardioverter defibrillator or heart transplantation [10,11,12]. A significantly elevated risk of in-hospital death was demonstrated among Takotsubo cardiomyopathy patients with versus without MB [13]. It seems reasonable to detect MB as early as possible to prevent a potential major event. MB has a male preponderance [14]. The prevalence of MB differs depending on the tools used for screening, ranging from 2% to 6% via invasive coronary angiography to 40% according to autopsies [7,15]. Although the number of reports on the topic is limited, MB with severe systolic luminal compression (SMB) is rare. Consequently, in some areas, patients with MB are prohibited from military service or training. The clinical presentations and exaggerating factors of MB mimic those of CAD [7]. Noninvasive evaluation modalities for ischemic heart disease, such as echocardiography, a treadmill test, or a nuclear medicine cardiology examination, are not effective in distinguishing between MB and CAD. To exclude the possibility of CAD, patients with MB and angina often have to undergo invasive coronary angiography or contrast coronary computed tomography (CT), both of which entail exposure to ionizing radiation [16]. So far, there is not a reliable, rapid, noninvasive, and cost-effective diagnostic method to identify the presence of myocardial bridging among anginal patients with low-risk profiles of coronary artery disease. Next-generation sequencing (NGS) has the potential to be a new alternative tool to CT and invasive angiography for rapidly and accurately diagnosing MB.

In the era of NGS, genetic tests are valuable tools for screening, diagnosis, and prognosis [17,18]. The mechanism of MB formation in terms of genetics and development has not been thoroughly investigated. Liu et al. investigated correlations between SNPs on the *APOE* gene and MB, but the results were not significant [19]. Currently, there is not a universal OMIM (Online Mendelian Inheritance in Man) identifier for MB. We explored potential underlying genetic variations associated with MB and a possible pathophysiology to understand the genetic factors surrounding MB and gain insight into interactions between myocardial vascular and cardiomyocytes (Figure 1).

## 2. Results

### 2.1. Variants Detected in Patients with SMB

Eight patients with SMB (mean age: 48.5 years, men: 50%) with prominent loss of the systolic luminal area were matched with eight controls for genetic sequencing (Appendix A). A total of 3,559,047 variants were detected among 39,577 genes in our study cohort (Figure 2). Nonsynonymous single-nucleotide variants and indels were selected, and we condensed the results to 21,334 variants in 8548 genes. Rare variants with allelic frequencies of <1% were selected from the Taiwan Biobank (TWB), the 1000 Genomes Project (1KGP), the Genome Aggregation Database (gnomAD), and the Exome Aggregation Consortium (ExAC) database, which yielded 3297 variants in 2530 genes. Variants or indels detected in the control group were excluded. A total of 56 pathogenic variants (including 49 coding variants) in 51 genes and 87 pathogenic variants (including 86 coding variants) in 85 genes for SMB were identified from ClinVar and CADD/REVEL, respectively. A detailed list of all the rare variants potentially pathogenic for SMB was provided in Appendix A.

### 2.2. Genes with Multiple Rare Pathogenic Variants in Patients with SMB

Figure 3 presents the distributions of rare pathogenic variants for SMB. Most candidate genes only included one variant. Although not being simultaneously predicted by both prediction systems, some genes included two or more rare variants for SMB. Genes with multiple rare variants pathogenic for SMB were also presumably important. The genes with two or more rare pathogenic variants were *BRCA2*, *DMD*, *TTN*, *FLNB*, and *SCN1A* (Table 1).

### 2.3. Rare Pathogenic Variants Identified by Both ClinVar and CADD/REVEL

In order to narrow down the gene list and to find the more important genes for SMB, we compared genes that appeared in both ClinVar and CADD/REVEL. Rare variants that were predicted to be pathogenic for SMB by both prediction systems were presumably important. Four rare pathogenic variants in four genes were identified by both ClinVar and CADD/REVEL, namely *SCN1A*, *PCDH15*, *SMAD9*, and *SGCA* (Table 2).

### 2.4. Tissue Expression Levels of Rare Variants Potentially Pathogenic for SMB

Genes with rare variants pathogenic for SMB from Table 1 and Table 2 were further examined according to the expression levels among cardiovascular and myocardial tissues. The Genotype-Tissue Expression (GTEx) Project was used to further investigate the expression of the genes harboring rare pathogenic variants in muscle and artery tissues. *TTN*, *DMD*, and *SGCA* exhibited high expression levels in the muscle and heart tissues (Figure 4). *DMD* and *SGCA* were also highly expressed in the artery tissues.

### 2.5. Functional Annotation of Rare Variants for SMB

We detected 132 SMB candidate genes by intersecting rare pathogenic variants identified by ClinVar and CADD/REVEL for enrichment analysis. The 132 SMB candidate genes were mapped to 124 mammalian (mouse) genes that were significantly enriched into 10 knockout mouse phenotypes, including abnormal cardiovascular system morphology, abnormal skeletal muscle morphology, abnormal skeletal muscle mass, abnormal muscle fiber morphology, and impaired skeletal muscle contractility. Similarly, the SMB candidate genes were enriched in the Kyoto Encyclopedia of Genes and Genomes (KEGG) pathway of arrhythmogenic right ventricular cardiomyopathy. Gene ontology (GO) was used to evaluate the molecular functions, cellular components, and the biological processes involved in SMB gene sets. A total of 10 GO terms were significantly enriched, including muscle contraction processes, the detection of mechanical stimuli, and ion transport systems (Table 3). The comprehensively detailed gene list of each category of gene set from functional annotation is shown in the Appendix A.

## 3. Discussion

The genetic basis of MB remains unclear. Here, we particularly focused on severe MB. In this study, we identified several pathogenic gene variants associated with SMB. Based on protein function, tissue expression levels, and the ClinVar and CADD/REVEL predictions, *TTN*, *DMD*, and *SGCA* were the highest-potential genes for SMB. All three genes are physiologically and morphologically essential to the cardiovascular system. Thus, we hypothesized that structural instability between cardiomyocytes and the surrounding connective tissues is critical for SMB.

*TTN* translates to the giant titin protein, previously known as connectin [20]. Titin is a large cytoplasmic protein in the human body that constitutes the backbone of the basic unit of cardiomyocytes: sarcomeres, which act as the fundamental driving force of heart power [21,22,23]. *TTN* has high expression levels in the heart and artery systems (Figure 4), the missense variants of which cause dilated cardiomyopathy [24,25,26,27]. As titin locates in the cytoplasm of cardiomyocytes with indirect attachment to the cell membrane via actin [28], genetic variants on *TTN* and subsequent derangement of titin might have a role in the formation of SMB under the interplay with anchor proteins on cell membrane, for example, the dystroglycan complex and sarcoglycan complex. In the current study, three rare variants were detected on *TTN* that only presented in the SMB patients, suggesting that *TTN* is a potentially important key to SMB. However, there could be another possibility that variants had a higher chance to be detected on *TTN* owing to the size of the giant gene. Due to the extreme abundance of titin in cardiomyocytes, a tiny change in the protein conformation might also translate to a tremendous structural impact and organ development. Also, due to the proximity of sarcomere units to the plasma membrane of cardiomyocytes, the effect of *TTN* variants on the formation of SMB currently could not be excluded, necessitating further evidence to address this issue.

*DMD* transcribes and translates to an essential protein on the muscular cell membrane: dystrophin [29]. The dystrophin-associated protein complex (DAPC) functions as an anchor protein, stabilizing the cell membrane of cardiomyocytes via linking in between the intracellular actin and the surrounding extracellular matrix (ECM; Figure 5) [30]. For decades, the majority of *DMD* research focused on muscular anomalies. Gene mutations of *DMD* caused several kinds of dystrophinopathies, including two rare muscular diseases: Duchenne muscular dystrophy and Becker muscular dystrophy [31,32]. Dystrophinopathies also manifest as dilated cardiomyopathies, which cause affected patients to develop progressive heart failure; newer treatments for dystrophinopathies are being extensively investigated [33]. The current study detected three genetic variants on *DMD* in three separate SMB patients, which might suggest an essential role of *DMD* in SMB. So far, evidence regarding the linkage between *DMD* and SMB is still lacking. The research field discussing the constitutional role in cell membrane assembly and mechanotransduction function of dystrophin is emerging in order to unravel the fundamental role of DMD in morphogenesis and basement membranes [34]. Based on the medical records, none of our enrolled patients with SMB exhibited relevant muscular diseases. DAPC is a laminin- and actin-binding glycoprotein that provides a robust connection between the cell membrane of cardiomyocytes and the surrounding connective tissues. We proposed that a loosened assembly of the cardiac muscles would enable the coronary arteries to penetrate through cardiomyocytes, leading to SMB. It was generally accepted that males had a higher prevalence for MB. Since the *DMD* locates on the sex chromosome, it seemed reasonable that males were more likely to be affected if the genetic variant for MB is located on chromosome X.

*SGCA* encodes a protein called “sarcoglycan-alpha”, a component of the dystrophin–glycoprotein complex that stabilizes cell membranes to the ECM [35]. Sarcoglycan alpha is one of the major sarcoglycan proteins assembled on the plasma membrane of striatal muscle fibers, including cardiomyocytes and skeletal muscle. Sarcoglycanopathy is an evolving science regarding a rare autosomal recessive hereditary muscular dystrophy affecting limb girdle muscles [36]. The sarcoglycan complex, along with the dystroglycan complex, functions as a connector between the cytoskeleton of the muscle fiber and the ECM, augmenting mechanical support during muscle contraction [37]. *SGCA* is highly expressed in striated muscles, and the heart has the highest expression levels (Figure 4). Because protein products from *SGCA* account for the structural stability between cardiomyocytes and the surrounding connective tissues [38], deformed *SGCA* products might result in a loosened myocardial mass structure and, thus, a tendency of coronary vasculature deviation into the myocardium. An animal experiment was once performed using Sgca-null mice for the characterization of cardiac structure and global functions, from which a significantly thickened interventricular septum and posterior wall, accompanied by the dilated left ventricular chamber, were recognized among Sgca-null mice [39]. However, there is no angiographical characterization study for Sgca-null mice so far. An animal study indicated that *Sgcd*-null mice had irregularities of the coronary vessels [40]. However, the effect of *SGCA* on the cardiovascular system remains unknown. Sarcoglycans are integrated into the dystrophin glycoprotein complex (DGC) and function as the fundamental anchor between cardiomyocytes and the ECM [41]. Because animal models have demonstrated that the destabilization of DGC leads to membrane fragility and a loss of integrity [42,43,44], genetic variants of *SGCA* might have a role in SMB (Figure 5). Indeed, a further study to confirm these functional variants is required.

Based on the ClinVar database, CADD/REVEL scoring tools, the degrees of tissue expression, individual gene functions, and locations of each protein product, we suggested that rare variants in *TTN*, *DMD,* and *SGCA* would play critical roles in the pathogenesis of SMB. All three genes were also included in both the “abnormal cardiovascular system physiology” and “abnormal cardiovascular system morphology” categories during functional annotation (Appendix A). Although the variants in the proposed culprit genes were not observed in all patients with SMB, our findings at least provided a new research direction and opened up a new field for addressing the underlying pathophysiology of SMB.

There are limitations in this study. First, there were very few enrolled study subjects. Consequently, the power of prediction might be relatively low, and there might be some undetected pathogenic rare variants for SMB. However, since we do not know exactly what the underlying mechanisms of MB are, we could not estimate very precisely the total amount of genetic variants pathogenic for MB from the available evidence. It might be difficult to calculate the power of the current study as to the efficiency of detecting pathogenic variants. Second, a functional study with animal experiment validation for the genetic variants of *DMD* and *SGCA* in SMB was not provided. Third, a high CADD or REVEL score does not necessarily mean that the variant is pathogenic. Therefore, false-negative or false-positive mining of variants could not be fully excluded. Fourth, the present study only highlights protein-coding variants identified from whole-exome sequencing. Thus, there might be non-coding genome wide variants that play an important role in the regulation of protein-coding pathogenic SMB genes. Fifth, because we compared those with severe MB in contrast to those with purely normal vessels, not those with milder MBs, the reported variants could be related to general MB rather than only to severe MB. Collectively, a combination of a larger patient cohort and animal model using whole-genome sequencing data would be very helpful to address these questions in the future.

## 4. Materials and Methods

### 4.1. Participants

Participants were recruited from Taipei Medical University Hospital, and the study was approved by the hospital’s Institutional Review Board. A total of 248 individuals had undergone coronary angiography. Those who presented with systolic compression in their coronary arteries but did not have any evidence of atherosclerotic CAD were treated as patients with MB. Each patient’s diagnosis of MB was independently confirmed by two cardiologists. The severity of MB was determined on the basis of the reduction in the luminal area. Because of a lack of universal angiographic definitions of SMB, we defined SMB as cases of MB with a loss of the luminal area greater than 80% based on our clinical experience. Control participants were individuals with normal coronary arteries, as confirmed via coronary angiography. To identify key variants carried by patients with SMB but not healthy patients, eight patients with SMB and eight age- and sex-matched control patients were selected for analysis. All participants provided informed consent for human body research.

### 4.2. Sample Preparation, Whole-Exome Sequencing, and Bioinformatics Analysis

DNA samples from the eight patients with SMB and eight normal patients were extracted using the QIAamp DNA Blood Maxi Kit (Qiagen, Germantown, MD, USA). Whole-exome sequencing using white blood cells was performed via polymerase chain reaction (PCR) amplification and the Nextera Illumina platform. The GATK discovery pipeline was used for PCR duplicate removal, quality score recalibration, GRCh38 (hg38) reference genome alignment, and variant calling. The annotation of functional consequences, pathogenicity, and minor allele frequencies for each variant was performed using ANNOVAR. Filtering for rare nonsynonymous variants and indels was performed using R. Our hypothesis was that rare genetic variants would contribute to the development of SMB. For this reason, we were specifically interested in rare variants with minor allele frequencies less than 0.01 according to data from TWB [45], 1KGP (East Asian) [46], gnomAD (East Asian) [47], and ExAC (East Asian) [48].

### 4.3. Pathogenicity Prediction

The definition and criteria for pathogenic variants were as follows: (A) ClinVar (version 20210501), those with conflicting evidence of pathogenicity, those likely pathogenic, pathogenic, or those pathogenic/likely pathogenic; (B) CADD Phred score > 20, indicating inclusion in the top 1% of the most pathogenic variants [49,50,51]; and (C) REVEL raw score > 0.75, calculated on the basis of 13 independent tools, with a higher score indicating a greater likelihood that the variant causes disease [52]. Gene expression levels among human tissues were viewed through the GTEx Portal (https://gtexportal.org/home/ accessed on 1 April 2022). Please refer to Figure 2 for study protocol.

### 4.4. Functional Annotation

Functional annotation provides relevant cellular, structural, and physiologic functions of a given gene set. The function of SMB pathogenic genes was examined to gain insight into the underlying mechanism of how these genes could cause SMB. Three databases were used for functional annotation: a mammalian phenotype ontology database [53], KEGG [54], and GO [55,56]. Mammalian gene knockout database provides abundant evidence that could not be carried out in humans. Knockout mouse phenotyping was performed using the Mammalian Phenotype Ontology database, which provides valuable insight into the physiological role of a gene in humans by demonstrating the consequences of gene ablation in mice. KEGG is a well-known knowledge base comprising many major databases, including GENES, PATHWAY, and DISEASE. KEGG pathway analysis provides information for understanding the biological pathways and interactions of genes. The GO Consortium incorporates extensive scientific research and provides consistently updated genes and gene products. It is a powerful tool for annotating genes with functional information and provides data on biological processes, molecular function, and cellular components. Over-representation analysis based on these three databases was performed using WebGestalt (WEB-based Gene SeT AnaLysis Toolkit). For all analyses, we considered an FDR of <0.05 to be significant.

## 5. Conclusions

The results suggest the roles of rare variants on *TTN*, *DMD*, and *SGCA* in the pathogenesis of SMB (Figure 6). Subsequent studies should focus on the *TTN*, *SGCA*, and *DMD*, because defective proteins and downstream protein complexes, including sarcomere, sarcoglycan, and dystroglycan, may lead to altered anchoring of cardiomyocytes and misguided vasculogenesis.

## Figures and Tables

**Figure 1 jpm-13-01509-f001:**
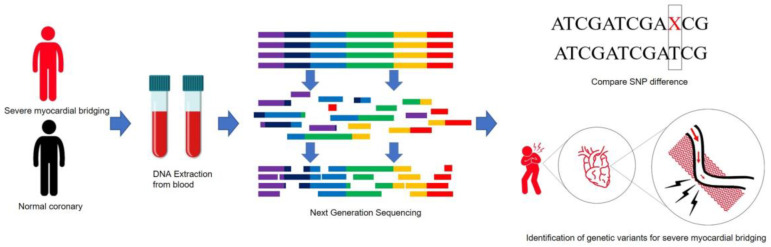
Flowchart of a genetic association study for SMB. Patients with MB might present with exertional chest pain. This is a case–control study that compared patients with SMB and healthy controls in terms of genetics, targeting hereditary differences etiologic for this coronary anomaly. SNP: single-nucleotide polymorphism. The “X” refers to any form of SNP.

**Figure 2 jpm-13-01509-f002:**
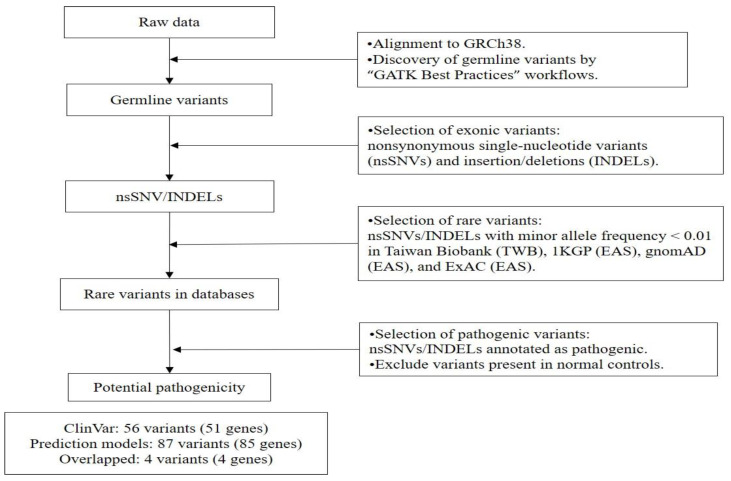
Gene variant exploration for SMB. 1KGP indicates the 1000 Genomes Project; EAS, East Asians; ExAC, Exome Aggregation Consortium; GATK, Genome Analysis Toolkit; gnomAD, The Genome Aggregation Database; GRCh38, Genome Reference Consortium Human Build 38 Organism; INDEL, insertion and deletion; SNV, single-nucleotide variants; TWB, Taiwan Biobank.

**Figure 3 jpm-13-01509-f003:**
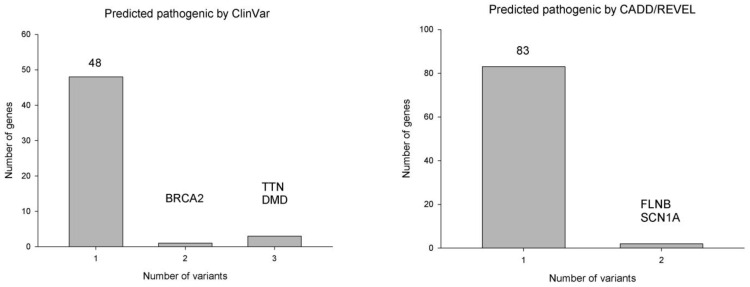
Rare pathogenic variants per gene for SMB predicted by ClinVar and CADD/REVEL.

**Figure 4 jpm-13-01509-f004:**
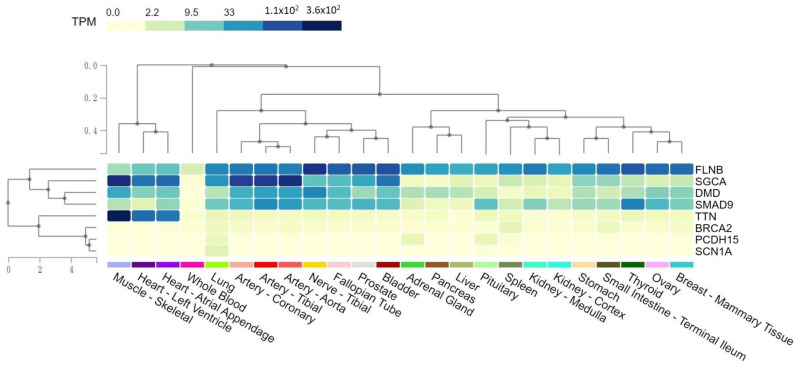
Tissue expression of genes potentially pathogenic for SMB.

**Figure 5 jpm-13-01509-f005:**
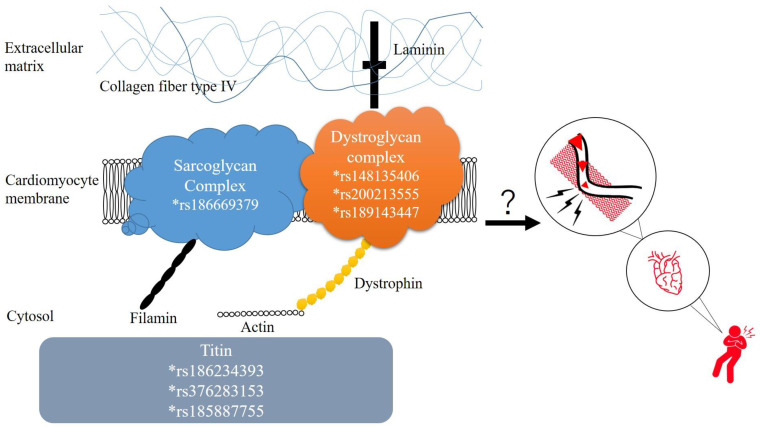
Interaction between the sarcoglycan complex (partially transcribed and translated from *SGCA*), the dystroglycan complex, and dystrophin (transcribed and translated from *DMD*). Sarcoglycans, dystroglycans, and dystrophins jointly construct the anchoring proteins on the plasma membrane of cardiomyocytes, augmenting fixation with the ECM. Our study suggests that variants of both genes might be correlated with SMB, which manifests as deviated coronary vessels into the myocardium. Also, *TTN* might have a role in SMB formation along with the cellular anchor proteins shown above.

**Figure 6 jpm-13-01509-f006:**
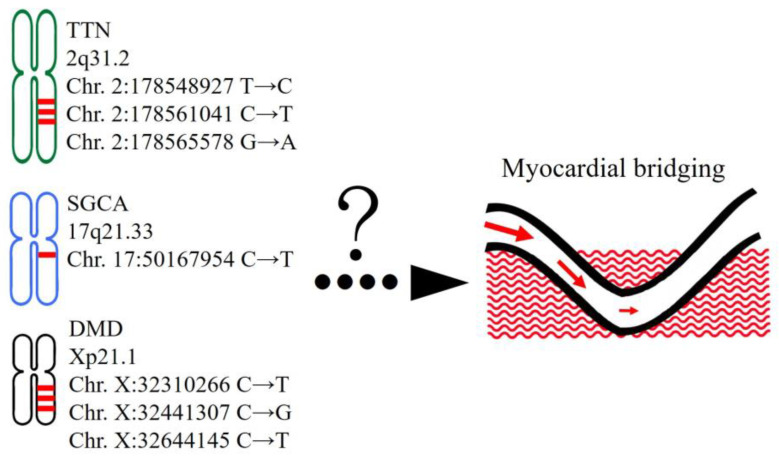
The postulated pathogenic rare variants for severe myocardial bridging from the current study were summarized with precise exon positions.

**Table 1 jpm-13-01509-t001:** Genes with multiple pathogenic variants for SMB predicted by ClinVar or CADD/REVEL.

Chr.	Position	Ref.	Alt.	Gene	Type	rs Number	ClinVar	CADD	REVEL	Number of Alt. Alleles in Patients with SMB							
										1	2	3	4	5	6	7	8
Pathogenic Variants Predicted by ClinVar																	
2	178548927	T	C	*TTN*	nsSNV	rs186234393	CIP	13.78	0.496	1	-	-	-	-	-	-	-
2	178561041	C	T	*TTN*	nsSNV	rs376283153	CIP	18.93	0.14	1	-	-	-	-	-	-	-
2	178565578	G	A	*TTN*	nsSNV	rs185887755	CIP	20.8	0.533	-	-	-	-	-	-	1	1
13	32338731	A	G	*BRCA2*	nsSNV	rs117187202	CIP	0.395	0.146	-	-	-	-	-	-	1	-
13	32340191	T	C	*BRCA2*	nsSNV	rs80358811	CIP	0.002	0.14	-	-	1	-	-	-	-	-
X	32310266	C	T	*DMD*	nsSNV	rs148135406	CIP	17.97	0.147	-	-	-	-	-	-	1	-
X	32441307	C	G	*DMD*	nsSNV	rs200213555	CIP	25.8	0.312	-	-	-	1	-	-	-	-
X	32644145	C	T	*DMD*	nsSNV	rs189143447	CIP	27.4	0.294	-	-	1	-	-	-	-	-
Pathogenic Variants Predicted by CADD/REVEL																	
2	165994164	C	T	*SCN1A*	nsSNV	rs121918808	LB	32	0.817	-	-	-	-	-	-	-	1
2	166041286	A	G	*SCN1A*	nsSNV	rs773695263	CIP	25.9	0.884	-	-	-	-	1	-	-	-
3	58008647	A	T	*FLNB*	nsSNV	N/A	N/A	32	0.953	-	-	1	-	-	-	-	-
3	58154853	C	T	*FLNB*	nsSNV	rs369477886	N/A	34	0.876	-	-	-	-	1	-	-	-

Chr., chromosome; Ref., reference allele; Alt., alternative allele; nsSNV, non-synonymous single nucleotide variant; CIP, Conflicting interpretations of pathogenicity; LB, likely benign; N/A, not applicable.

**Table 2 jpm-13-01509-t002:** Rare pathogenic variants predicted by both ClinVar and CADD/REVEL scores in patients with SMB.

Chr.	Position	Ref.	Alt.	Gene	Type	rs Number	SMB No.
2	166041286	A	G	*SCN1A*	nsSNV	rs773695263	5
10	53857257	C	T	*PCDH15*	nsSNV	rs201137087	4
13	36879563	T	C	*SMAD9*	nsSNV	rs397514715	6
17	50167954	C	T	*SGCA*	nsSNV	rs186669379	3

Ref., reference allele; Alt., alternative allele; SMB No., patient ID.

**Table 3 jpm-13-01509-t003:** Functional annotation of SMB candidate genes.

Functional Annotation	Reference Genes in Category	SMB Genes in Category	*p* Value	FDR
**Knockout mouse phenotype**				
Abnormal soleus morphology	21	4	3.84 × 10^−5^	2.50 × 10^−2^
Impaired skeletal muscle contractility	38	6	1.25 × 10^−6^	8.18 × 10^−3^
Absent startle reflex	39	5	2.91 × 10^−5^	2.22 × 10^−2^
Decreased skeletal muscle mass	107	8	6.75 × 10^−6^	1.85 × 10^−2^
Abnormal skeletal muscle mass	121	8	1.67 × 10^−5^	2.04 × 10^−2^
Abnormal muscle fiber morphology	322	13	8.84 × 10^−6^	1.85 × 10^−2^
Increased or absent threshold for auditory brainstem response	310	12	3.06 × 10^−5^	2.22 × 10^−2^
Abnormal skeletal muscle morphology	381	14	1.14 × 10^−5^	1.85 × 10^−2^
Abnormal cardiovascular system physiology	1421	29	2.56 × 10^−5^	2.22 × 10^−2^
Abnormal cardiovascular system morphology	1794	34	1.88 × 10^−5^	2.04 × 10^−2^
**KEGG pathway**				
Arrhythmogenic right ventricular cardiomyopathy	72	7	3.49 × 10^−6^	1.14 × 10^−3^
**GO term categories**				
Detection of mechanical stimulus	43	6	6.03 × 10^−7^	2.74 × 10^−3^
Muscle contraction	339	14	1.52 × 10^−7^	1.38 × 10^−3^
Muscle system process	423	14	2.15 × 10^−6^	3.33 × 10^−3^
Monovalent inorganic cation transport	513	15	4.17 × 10^−6^	4.21 × 10^−3^
Inorganic cation transmembrane transport	722	19	9.42 × 10^−7^	2.85 × 10^−3^
Cation transmembrane transport	810	20	1.27 × 10^−6^	2.88 × 10^−3^
Metal ion transport	841	20	2.25 × 10^−6^	3.33 × 10^−3^
Inorganic ion transmembrane transport	808	19	4.91 × 10^−6^	4.47 × 10^−3^
Cation transport	1111	23	3.73 × 10^−6^	4.22 × 10^−3^
Ion transport	1608	29	2.56 × 10^−6^	3.33 × 10^−3^

FDR, false-discovery rate; KEGG, Kyoto Encyclopedia of Genes and Genomes; GO, Gene Ontology.

## Data Availability

The clinical data were collected at Taipei Medical University Hospital. Raw data for sequencing were generated at Taipei Medical University. Data supporting the findings of the study are available from the corresponding author upon request.

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
