# Peer review of "Identification of Genetic Variants Associated with Severe Myocardial Bridging through Whole-Exome Sequencing"

_jpm, 2023, doi:10.3390/jpm13101509_

Round 1
Reviewer 1 Report
Overall this is a interesting study and promising data has been shown, while some of the analysis could be potentially improved as following:
1. line77, what is the detailed criteria of variants filtering?
2.Although Myocardial Bridging is a rare disease, it doesn’t necessarily mean that its potential variants have low frequency.
3.In fig.4, these genes are not seemed to be very tissue specific, how to explain it? Do these identified variants have any high LD or association with QTLs of these genes?
4.In addition to performing functional annotation of genes, is there any genetic association or high LD between identified variants and GWAS SNPs of cardiovascular diseases?
Author Response
Point-by-point Response to Reviewer’s Comment
Thank you very much for reviewing our research and giving us valuable comments for improvement.
Reviewer’s comment:
Overall this is a interesting study and promising data has been shown, while some of the analysis could be potentially improved as following:
- line77, what is the detailed criteria of variants filtering?
Response 1: Thank you for your comment.
We performed Whole-exome sequencing, from which all exome reads from MB and normal control were compared to human reference genome. Variants were filtered out via simple comparison to reference genome by GATK workflow. Synonymous variants were excluded. Then, we particularly selected genetic variants with minor allele frequencies < 1% from public database (TWB, 1KGP, gnomAD, and ExAC). Rare genetic variants presented in the normal controls were excluded from further analysis. Pathogenicity of obtained variants were analyzed by two commonly used prediction systems: ClinVar and CADD/REVEL.
Thank you again for your detailed reviewing.
- Although Myocardial Bridging is a rare disease, it doesn’t necessarily mean that its potential variants have low frequency.
Response 2: Thank you for your comment. Yes, indeed. There might be some potential variants with higher frequency that is associated with myocardial bridging. In order to find the most important gene variants for MB, we particularly picked MB with overwhelming severity. According to a review article from Journal of the American College of Cardiology (J Am Coll Cardiol 2021;78:2196–2212), the prevalence of MB from coronary angiography range from 2% to 6%. We had searched and collected extremely severe MB subjects with rigorous definitions. The prevalence of “severe” MB (SMB) from our cohort is 10% to 20% of all MB, which means that the overall prevalence of SMB is around 1%. That’s why we extracted rare variants with allele frequency < 1% in the methodology. In the future, we may broaden the search criteria for potential gene variants for general MB.
Thank you for your question and concern.
- In fig.4, these genes are not seemed to be very tissue specific, how to explain it? Do these identified variants have any high LD or association with QTLs of these genes?
Response 3: Thank you for your comment. Yes, these genes were not 100% specific to a specific organ, but, more or less, exhibited different levels of expression. Figure 4 helped us make clear which genes deserved more attention as to cardiovascular systems.
Since we had selected rare variants with minor allele frequency < 1%, the LD of our genes of interest were quite low.
Our study design had not quantification profiles of each gene on specific organ from our enrolled subjects, thus, we don’t have the data of QTLs of these genes.
Based on data from the GTExPortal, currently all the identified variants in this study had not had association with QTLs, including both the eQTL and sQTL.
Thank you again for your comment.
4.In addition to performing functional annotation of genes, is there any genetic association or high LD between identified variants and GWAS SNPs of cardiovascular diseases?
Response 4: Thank you for your comment. Because we selected “pure” MB subjects to compare with subjects having “completely normal” coronary arteries, we have not performed the cross-comparison between known GWAS SNPs of cardiovascular disease (mostly meant to be coronary artery disease (CAD)) and our genes of interest.
According to Journal of Human Genetics (2013) 58, 120-126, 3 GWAS SNPs, including 1p13.3/SORT1 (rs599839), 9p21.3/CDKN2A/2B (rs4977574), and 11q22.3/ PDGFD (rs974819), were associated with CAD among Europeans and East Asians. The identified variants of SMB in our study locate on Chromosome 2, 17, and X, none of which had the same chromosome as the known GWAS SNP. Thus, currently there is not available LD could be presented.
Indeed, in the future we will enroll subjects with both MB and CAD, from which the genetic association and LD might be calculated and evaluated.
Thank you again for your evaluable suggestions.

Reviewer 2 Report
Present study conducted by Shih et al. used a whole-exome sequencing technique and analyzed genotypic differences between eight coronary angiography–confirmed cases of severe MB and eight age- and-sex-matched control patients without severe MB. Among the gene candidates for severe MB, rare variants in DMD and SGCA were identified to be the most important for pathogenesis of severe MB. The study seems to be interesting. However, I have some major comments:
1. Authors have compared eight coronary angiography–confirmed cases of severe MB and eight age- and-sex-matched control patients without severe MB in current study. In supplement 1, in note column, they write healthy controls for patients without SMB which is not appropriate. They may choose to write patient with/ without SMB in sample column and may want to remove note column as that seems confusing.
2. How did authors choose the sample size of 8 patients and 8 controls for their study? In other words, it is recommended to calculate the power of the study to effectively detect all pathogenic variants related to severe myocardial bridging.
3. It is not clear how the authors highlight only DMD and SGCA genes for SMB pathogenesis as the most crucial in abstract? Since TTN also seem to express highly in muscle and heart tissues and supported based on protein function and the ClinVar and CADD/REVEL predictions and can be highlighted in abstract. This comment is also consistent with figure 5 in the manuscript where authors preferentially show SNPs of DMD and SGCA but not for TTN? Functions of TTN gene with its associated SNPs can be also highlighted in the figure 5.
4. SMB Gene names involved in each pathway described in Table 3 can be given in supplementary to make this table information more useful for readers.
5. It is recommended to add a figure where exonic locations of identified pathogenic variants in lead shorlisted SMB genes (DMD, SGCA and TTN) can be shown.
6. The present study only highlights protein-coding variants identified using whole-exome sequencing. However, there can be non-coding genome wide variants that play an important role in regulation of protein-coding pathogenic SMB genes. Authors can certainly add this as limitation of their study where future studies with larger sample sizes covering whole genome sequencing data for patients with SMB vs. patients without SMB will be more useful to answer such questions.
Author Response
Point-by-point Response to Reviewer’s Comment
Thank you very much for reviewing our research and giving us valuable comments for improvement.
Reviewer’s comment:
Present study conducted by Shih et al. used a whole-exome sequencing technique and analyzed genotypic differences between eight coronary angiography–confirmed cases of severe MB and eight age- and-sex-matched control patients without severe MB. Among the gene candidates for severe MB, rare variants in DMD and SGCA were identified to be the most important for pathogenesis of severe MB. The study seems to be interesting. However, I have some major comments:
- Authors have compared eight coronary angiography–confirmed cases of severe MB and eight age- and-sex-matched control patients without severe MB in current study. In supplement 1, in note column, they write healthy controls for patients without SMB which is not appropriate. They may choose to write patient with/ without SMB in sample column and may want to remove note column as that seems confusing.
Response 1: Thank you for your suggestion. We had refined our Supplement 1 to a more readable form according to your recommendation. It really looks more understandable.
Thank you very much.
- How did authors choose the sample size of 8 patients and 8 controls for their study? In other words, it is recommended to calculate the power of the study to effectively detect all pathogenic variants related to severe myocardial bridging.
Response 2: Thank you for your comment. What we observed were patients with extremely severe MB patients, and it was not easy to enroll and collect them. We used whole exome sequencing technique for its higher resolution of sequencing, by which we utilized for searching variants only in SMB patients. We would like to add this concern into the limitation section.
Thank you again for your recommendation.
- It is not clear how the authors highlight only DMD and SGCA genes for SMB pathogenesis as the most crucial in abstract? Since TTN also seem to express highly in muscle and heart tissues and supported based on protein function and the ClinVar and CADD/REVEL predictions and can be highlighted in abstract. This comment is also consistent with figure 5 in the manuscript where authors preferentially show SNPs of DMD and SGCA but not for TTN? Functions of TTN gene with its associated SNPs can be also highlighted in the figure 5.
Response 3: Thank you for your comment. We particularly highlighted the role of SGCA and DMD because both protein products locate on the plasma membrane of cardiomyocytes, in charge of cellular anchoring to the surrounding connective tissues. The protein product of TTN, titin, locates in the intracellular space without direct contact to extracellular area, so we postulated that TTN might have lesser role in myocardial bridging. We still incorporated the TTN genetic variants into the Figure 5 and it looked much clearer.
Thank you again for your suggestions.
- SMB Gene names involved in each pathway described in Table 3 can be given in supplementary to make this table information more useful for readers.
Response 4: Thank you for our suggestion. We had managed to list the table of detailed genes in each individual gene set of functional annotations, as shown in Supplement 3. Indeed, it provides more information for readers.
Thank you again for your valuable recommendation.
- It is recommended to add a figure where exonic locations of identified pathogenic variants in lead shortlisted SMB genes (DMD, SGCA and TTN) can be shown.
Response 5: Thank you for your suggestion. It is a good idea that we show a figure with precise position of each variant of our shortlisted genes. We thus added a Figure 6 to briefly summarize the locations of our pathogenic variants of interest for SMB.
Thank you again for your recommendation.
- The present study only highlights protein-coding variants identified using whole-exome sequencing. However, there can be non-coding genome wide variants that play an important role in regulation of protein-coding pathogenic SMB genes. Authors can certainly add this as limitation of their study where future studies with larger sample sizes covering whole genome sequencing data for patients with SMB vs. patients without SMB will be more useful to answer such questions.
Response 6: Thank you very much for your suggestion. Our exon-only data indeed increased the chance of missing non-protein-coding variants in SMB patients. We had added this as limitation in our manuscript. We will try to proceed to whole genome sequencing in the future if feasible.
Thank you again for your comment.

Round 2
Reviewer 1 Report
The manuscript has been improved, I have no further questions.
Author Response
Thank you very much for your valuable suggestions that improved our study.
Reviewer 2 Report
Authors have significantly improved the manuscript. However, I still have some major comments:
1. Authors have identified genetic variants for severe myocardial bridging (SMB) by comparing SMB patients with normal controls. However, the correct control group for such studies should be patients with myocardial bridging that is not severe (cases of MB with a loss of the luminal area less than 80% based on clinical experience. Such comparisons will identify true positive genetic variants associated with severe cases of MB. In present study, some of the reported variants could be related to MB rather than SMB. It is recommended to add this as one of the limitations of the study.
2. Authors does not seem to state the genome version of the coordinates of identified variants in analyses methods i.e., hg19 or hg38.
3. Is there any sex specific role of DMD that have been observed in literature since it is in chromosome X? Is it possible that the variants in this gene may have sex-specific predisposition for SMB/MB?
Author Response
Point-by-point Response to Reviewer’s Comment
Thank you very much for reviewing our research and giving us valuable comments for improvement.
Reviewer’s comment:
Authors have significantly improved the manuscript. However, I still have some major comments:
- Authors have identified genetic variants for severe myocardial bridging (SMB) by comparing SMB patients with normal controls. However, the correct control group for such studies should be patients with myocardial bridging that is not severe (cases of MB with a loss of the luminal area less than 80% based on clinical experience. Such comparisons will identify true positive genetic variants associated with severe cases of MB. In present study, some of the reported variants could be related to MB rather than SMB. It is recommended to add this as one of the limitations of the study.
Response 1: Thank you for your suggestion. Exactly, we could not exclude the possibility that the reported variants could be related to MB rather than only to SMB. Since the severity of MB presented as a continuum spectrum (from very mild to very severe), we strategically chose to select the extreme ends of MB severity (from purely normal vessel to maximally severe MB, which was relatively rare), trying to yield the most relevant genetic differences in between.
We added this concern to the Limitation Section of the manuscript. (page 9, line 262)
Thank you again for your valuable suggestion.
- Authors does not seem to state the genome version of the coordinates of identified variants in analyses methods i.e., hg19 or hg38.
Response 2: Thank you for your reminder. Within the GATK discovery pipeline, we utilized the GRCh38 for genome alignment. We added the synonym “hg38” behind the GRCh38 in the manuscript. (page 9, line 287)
Thank you again for your reminder.
- Is there any sex specific role of DMD that have been observed in literature since it is in chromosome X? Is it possible that the variants in this gene may have sex-specific predisposition for SMB/MB?
Response 3: Thank you for your comment. Yes! It is an important concern that gender difference should be mentioned if the reported genetic variants are located on chromosome X. According to a recent meta-analysis (Clinical Anatomy. 2021;34: 685 – 709), MB has a male preponderance rather than a female. Although males were more likely to have coronary angiography, it was generally accepted that males had a higher prevalence of MB. It seemed reasonable that if the genetic variant for MB was located on chromosome X, males were more likely to be affected.
These concepts were added to the manuscript in the Introduction (page 2, line 63) and Discussion (page 7, line 204).
Truly thank you very much for reminding us of this important issue.
